# High voltage DC power supply with power factor correction based on LLC resonant converter

**Muhammad Abid[1], Fiaz Ahmad[2], Farman Ullah[1], Usman Habib[3], Saeed Nawaz[1], Mohsin Iqbal[3]\*, Ajmal Farooq[4]**

**1** Department of Electrical Engineering, COMSATS Institute of Information Technology, Attock, Pakistan,
**2** Department of Electrical & Computer Engineering, Air University, Islamabad, Pakistan, **3** Department of Electrical Engineering, National University of Computer & Emerging Sciences, Islamabad, Pakistan,
**4** Department of Electrical Engineering, Sarhad University of Science & IT, Peshawar, Pakistan

\* Engr.mohsiniqbal08@gmail.com

**Data Availability Statement:** All relevant data are within the manuscript and its Supporting Information files.

## Abstract

This paper presents analysis, design and experimentation of a high voltage DC power supply (HVDCPS) with power factor correction based on LLC resonant converter. For power factor correction improvement, the proposed topology has an input rectifier with two filter capacitors, two inductors with a bus capacitor (Cbus) and a resonant tank. To prevent the reverse current flow towards the source diodes (D9 & D10) are employed. A couple of power switches are inserted in a single leg that makes a half-bridge network. To form an LLC resonance circuit, a capacitor and two inductors are connected to the primary winding of the high voltage transformer (HVT). To rectify the high frequency and high voltage, a full-bridge rectifier is inserted to secondary side of high voltage transformer (HVT). The secondary diodes always get on and off under zero current switching (ZCS) due to discontinuous conduction mode of proposed topology. It is found that due to power factor correction, less cost, lower losses and smaller size, the proposed topology achieves several major improvements over the conventional high voltage power supply. To obtain zero voltage switching (ZVS) the converter operate in a narrow frequency range. The output voltage can be varied or regulate through pulse width modulation of power switches. Due to ZVS and ZCS, the proposed topology has minimum switching losses and therefore higher efficiency. To verify the feasibility of the proposed topology a prototype is being implemented and verified by simulation & experimental results for 1.5KV prototype of the proposed topology. The results make sure the achievement, good efficiency and successful operation of the proposed topology.

## Introduction

High-voltage DC power supplies are vastly implemented in particle accelerators, electrostatic precipitation, and high voltage pulse generators, focused ion of beam columns, laser, X-ray systems, electron microscopes and also employed in a many of other applications like in

**Funding:** The author(s) received no specific funding for this work.

**Competing interests:** All the authors hereby declare that no competing interests exists regarding this manuscript.

industries, in scientific research and testing laboratories. Some of the HVDCPS's generate analog input, which is usable in control of output voltage [1–6].

Moreover, high voltage DC generator contains few other applications in different fields like filtration of various gases and dust, electrostatic painting or coating and precipitation. These type of mechanisms need very larger level voltage for the proper processes [7].

Describing the structure and construction, a high voltage DC power supply consists of a diode rectifier with a filtering capacitor, an inverter with larger frequency, a high frequency and high voltage transformer (HVT), and a controller and filtering capacitor used with high voltage rectifier [8–10]. Among these components, HVT is much complicated part and it has a huge impact on the power supply's performance. As it contains higher turn ratio, so to harvest the effectual suitable insulation voltage between the primary winding and secondary winding, a sufficient geometrical distance is essential. These insulation requirements exasperate transformer non-idealities, which include parasitic capacitance, and leakage inductance that causes the spikes of current and voltage, and increase noise and losses [11–14].

For the utilization of these non-idealities as utile elements, there are several types of converter such as parallel resonant converters (PRCs), series-parallel resonant converters (SPRCs) and series resonant converters (SRCs) had been suggested by the earlier researchers [15, 16].

One of these converters, is LLC resonant converter that is most useful converter topology due to its simple structure [17–20] and has many other advantages as well. It is free of saturation of transformer, allows capacitive output-filter, zero voltage switching, zero current switching, less switching losses at higher frequencies, can also absorb the transformer's leakage inductance [21]. LLC resonant converter has much reduced losses and fewer problems compared to the parallel resonant converter (SRC) and series-parallel resonant converter (SPRC), due to sinusoidal behavior of LLC resonant converter. Therefore, LLC SRC got huge attentions in last few years due to its good operational characteristics and higher frequency for operation range.

In the proposed resonant topology, the LLC resonant converter is designed to operate below the resonance in order to harvest zero voltage switching (ZVS) [22]. The output diodes of secondary bridge rectifier are turned-on and off under zero current switching (ZCS) in this operating region which decreases the switching losses [17, 23–25]. Due to ZCS and ZVS, a well improved power factor is obtained and hence the overall efficiency of the proposed topology has improved. To employ constant output voltage applications, various authors have been analyzing the LLC resonant converter.

In proposed paper, a specific design for a high voltage DC power supply with power factor correction using half-bridge resonant converter is proposed that can generate an output voltage of 1.5KV. The simulation of the proposed circuit is tested by using simulating tool called PSPICE. The amplitude of output voltage is being regulated by changing the frequency of pulse width modulation (PWM), that is applied at gate terminal of MOSFETs'.

## Description and working principle

The proposed system for designing a high voltage DC power supply with power factor correction topology can be elaborated by circuit diagram as shown in Fig 1. The proposed design is composed of an input rectifier (D1- D4) with two filter capacitors (C1 & C2), two inductors (Lf & Lb), two diodes (D9 & D10), a bus capacitor (Cbus), a larger frequency inverter with LLC resonant tank, a higher frequency and high voltage transformer (HVT), a high voltage secondary rectifier having an output filter capacitor Co, and a resistive load connected at output. Here, the leakage inductance Lm of HVT is unified as a useful element. The Cds1and Cds2 is the output parasitic capacitances of power switches S1 and S2, respectively. Power

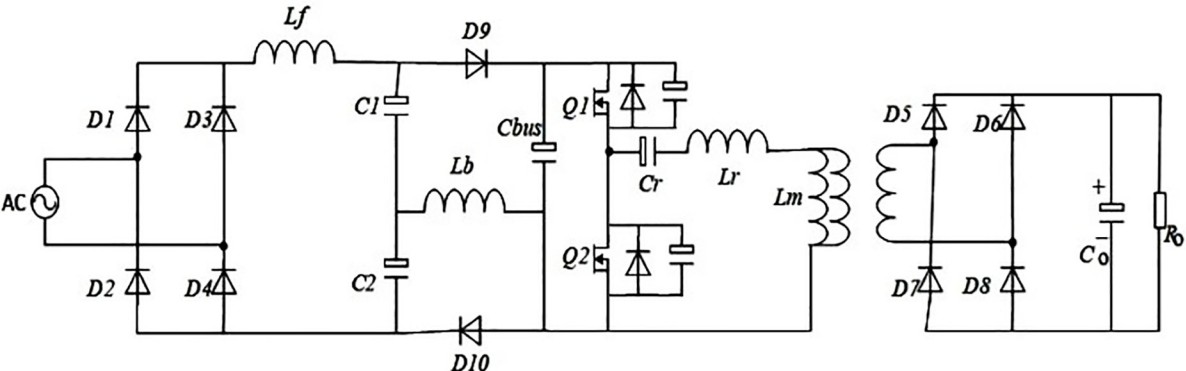

**Fig 1. High voltage DC power supply with power factor correction.**

switches S1and S2 are connected in a single leg form half-bridge inverter. The input rectifier is connected to the bus capacitor (Cbus) via two filter capacitors (C1 & C2), one inductor Lf and two diodes (D9 & D10). The power MOSFETs are connected across the primary windings of the transformer via resonance capacitor Cr, magnetizing inductor Lm and resonance inductor Lr. The formation of resonant tank is carried-out by arranging Cr, Lr and Lm in the described way. At the secondary side of HVT, a high voltage full bridge rectifier (D5-D8) and the output resistive load are connected in series via filter capacitor Co.

The half bridge inverter generates a square wave of voltage by switching with the alternation of approximately 50% duty cycle, which is exactly for the equivalent time. Actually, a small dead time is inserted between instants of turning on and off, the power switches (Q1 and Q2). This dead time is very necessary for the better operation of converter. It is very sure that there will be no cross conduction between power switches Q1 and Q2 and in the next section it will be clarified as well. This dead time allows the ZVS to be achieved. The Cbus capacitor employed as an input DC source for switching network that converts the DC input voltage into square wave, having an amplitude equals to Vin and a fixed duty cycle.

## Steady-state analysis

The complete steady-state waveforms of designed converter are shown in Fig 3. The complete operational cycle of proposed converter is divided into eight modes of operations. Here, only first half cycle of operation is elaborated because the operation of the next half cycle is also symmetrical to the first one. The equivalent circuit shown in Fig 2 expresses each mode of operation accordingly.

### Mode 1 (t1- t2)

This stage begins when the voltage across Q1is reduced to vanishing point before it turns on. Therefore, resonance current (iLr) begins to flow through the body diode of Q1 in positive direction, in sinusoidal manners. Due to the discharging of resonant capacitor Cr, resonance current (iLr), rises and the energy flows towards the source. The switch S1 recognizes the ignition of the zero current because the resonant currents start to go up from scratch. The magnetizing current, iLr, will also enhance linearly from negative towards positive. The secondary rectifier diodes D5and D8 turn on under zero current switching condition and transfer energy towards the load (Ro), through the filter capacitor. This mode of operation is shown in Fig 2

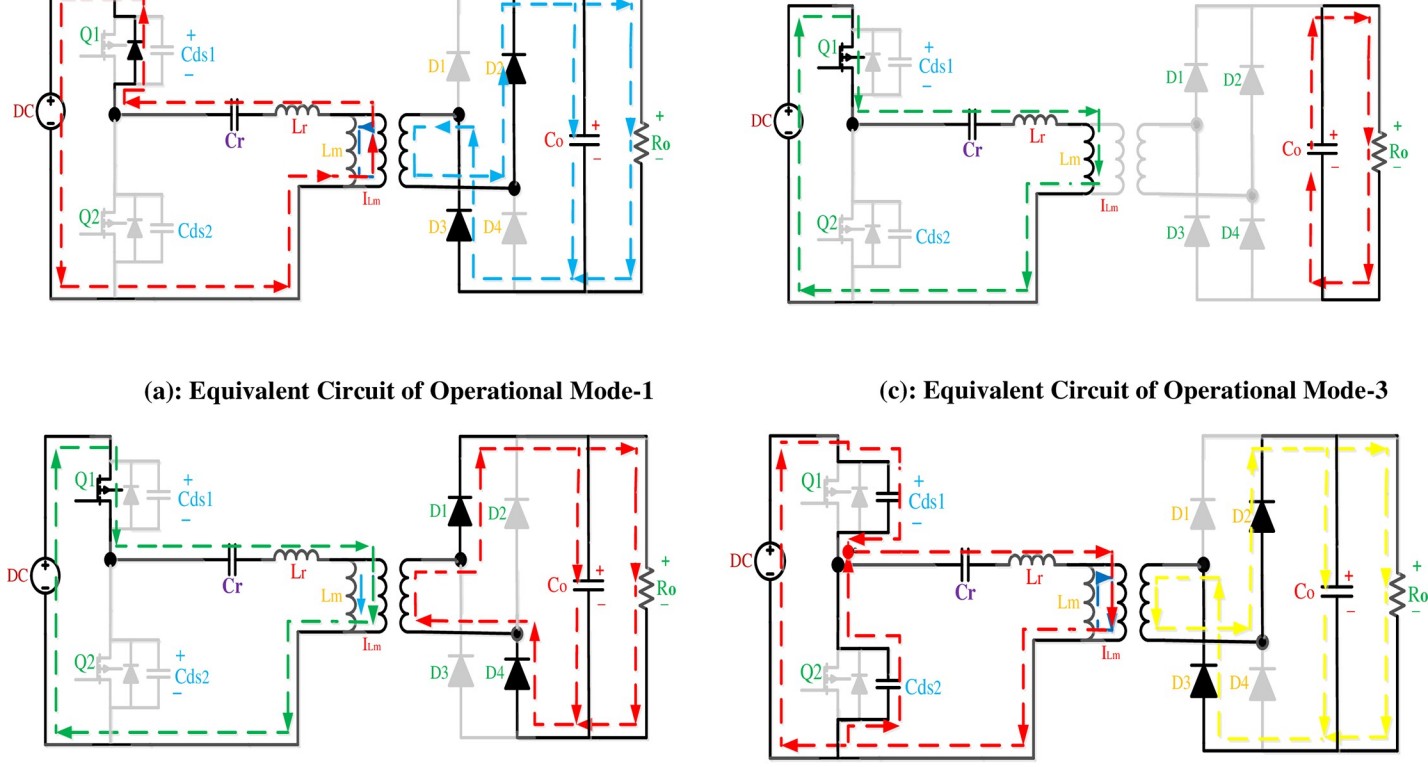

**(a): Equivalent Circuit of Operational Mode-1**

**(c): Equivalent Circuit of Operational Mode-3**

**(b): Equivalent Circuit of Operational Mode-2**

**(d): Equivalent Circuit of Operational Mode-4**

**Fig 2.** **(a)**: Equivalent circuit of operational mode-1. **(b)**: Equivalent circuit of operational mode-2. **(c)**: Equivalent circuit of operational mode-3. **(d)**: Equivalent Circuit of Operational Mode-4.

(A). At t = t0, the resonance currents iLr becomes zero and that is the instant of ending the Mode-1. By defining ILm0,Vcr0 and ILr0 as the initial condition at Wr = $2\pi^*$ fr and t = t0.

## Mode 2 (t2- t3)

In the start of this stage, Q1 is turned on and the resonant current (iLr), becomes positive. Input voltage source charges Lr and Cr, and there is a net positive flow of energy to the output load. The secondary rectifier diodes D5and D8 are still conduct under ZCS during this mode and deliver energy to the load. The magnetizing inductor Lm charged linearly due to resonance between Cr and Lr and is still clamped to nVo. During this mode the magnetizing inductor Lr does not participate in resonance. The state at which iLr becomes equal to iLm, and the output current is vanished, is the end of the operational mode-2.

## Mode 3 (t2- t3)

This mode starts with zero current flowing towards the primary winding of the transformer, because, iLm, becomes equal to iLr. The magnetizing inductance, Lm, and resonance inductance is now in series and both of them are active in resonance operation with Cr. As the output is isolated exists separately from input, so producing no power at the secondary side. The secondary diodes are turned off under zero current switching condition during this mode. As iLr and iLm same in magnitude, so the current travels regularly in the primary side only. Now,

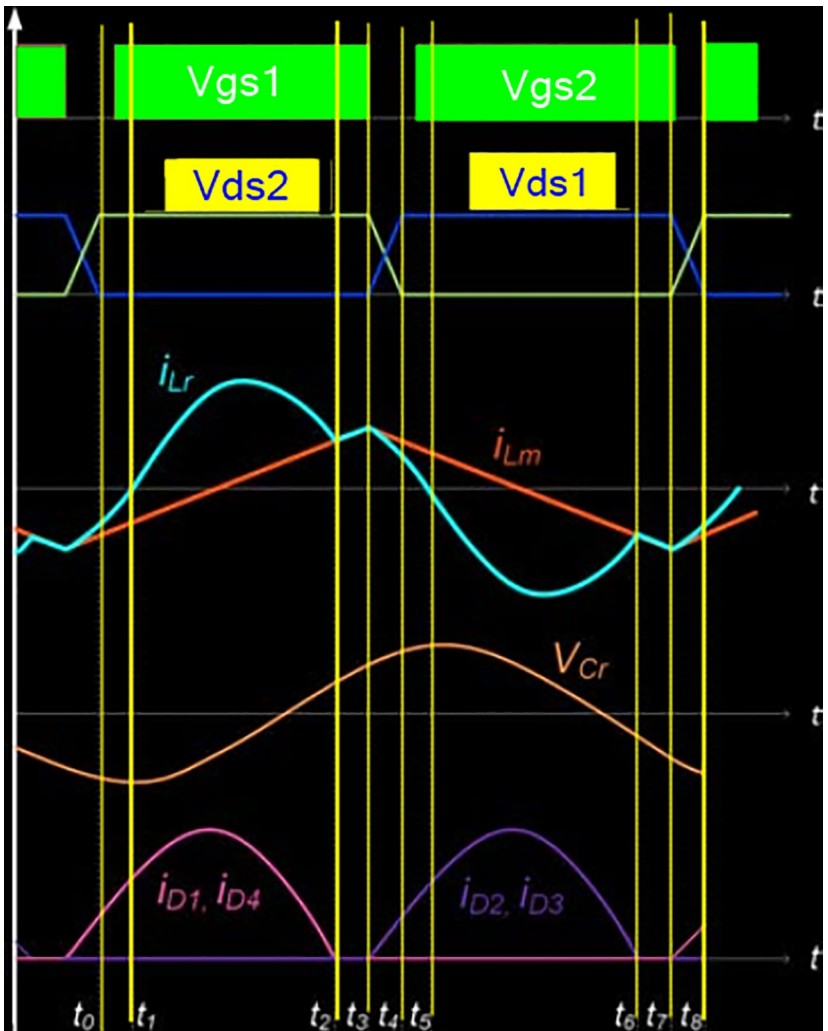

**Fig 3. Steady-state waveforms of LLC resonant converter.**

the equivalent inductance active in resonance boosts from Lr to get an increment of Lm (Lr + Lm), while the currents ILm and ILr are nearly equal in this minor period. This mode terminates by turning off the power switch Q1 and voltage across it begins rising.

### Mode 4 (t3- t4)

At the starting of this stage power switches Q1 andQ2 are turned-off. The resonant current, ILr, is flowing through output parasitic capacitors of Q1 andQ2, charging and discharging Cds2 and Cds1, respectively. During this mode the magnetizing current, ILm, begins to decrease linearly and Lm, is clamped to a level of voltage -nV0. Lm no longer participate in resonance. The mode end when the power diode of Q2 is forward-biased and begins to conduct. These four modes complete the first half-cycle of the operation.

### Gain characteristics of LLC resonant converter

It is well known to all, the most widely employed and effectual technique to overcome the impacts of distributed parameters is LLC resonant technique. In the above section, the voltage gain characteristics of the proposed system are analyzed.

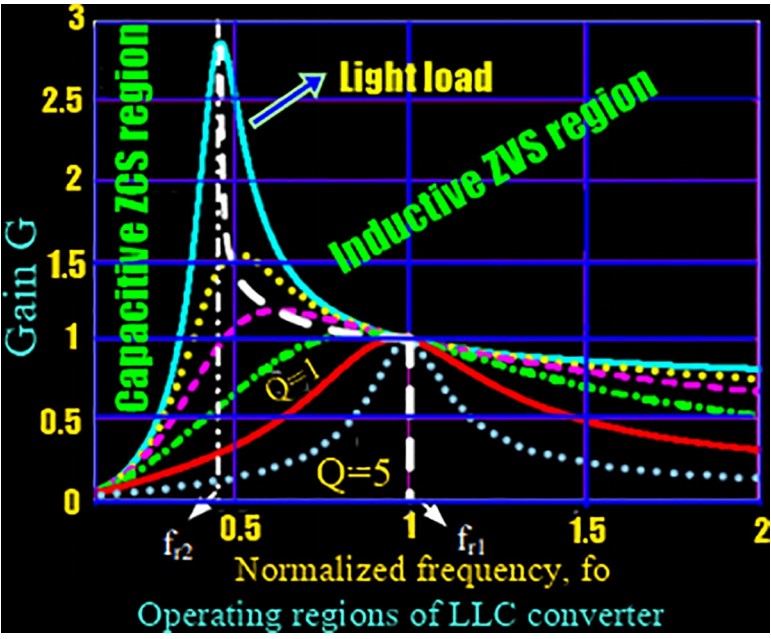

**Fig 4. Operating regions of LLC converter.**

The Fig 4 given below shows that all gain curves have peaks which define the limit between capacitive and inductive impedances of the LLC resonant tank, so we can define the capacitive and inductive operation regions as shaded in the figure. As the zero-voltage switching (ZVS) is achieved only in inductive region, therefore, it is desired to hold an inductive operation across the load current ranges and entire input voltages, and never fall into the region of capacitive operation. In addition to that, during the capacitive operation the current leads the voltage, so that before the MOSFET turns-off the current in the MOSFET will reverse direction, then the reverse current will flow in the body diode of the power MOSFET after the power MOSFET turn- off, which will causing hard commutation of a body diode and also cause noise and reverse recovery losses and might cause device failure and high current spikes.

In Fig 4, it is shown that high Q curves belong to heavier loads while the low curves represent lighter load operation. It also shows that all Q curves have a unity gain (load conditions) and cross at the resonant frequency point (at Fx = 1 or fs = fr).

Converter gain = Switching Bridge Gain * Resonant Tank Gain * Transformer Turn Ratio (Ns/Np).

The values of switching bridge gains for half bridge and full bridge are 0.5 and unity, respectively. LLC resonant tank gain can be derived by the analysis of resonant equivalent circuit (see, Fig 5). Mathematically, the magnitude of the resonant tank gain is equal to the magnitude of its transfer function as expressed in Eq 1.

$$K(Q, m, Fx) = \frac{Fx^2(m-1)}{\sqrt{(m.Fx^2 - 1)^2 + Fx^2.(Fx^2 - 1)^2.(m-1)^2.Q^2}} \tag{1}$$

The LLC resonant switching frequency $fs$ should must lies in the range ($f_m, f_r$) for the proper operation of switches and secondary side diodes in ZVS and ZCS respectively.

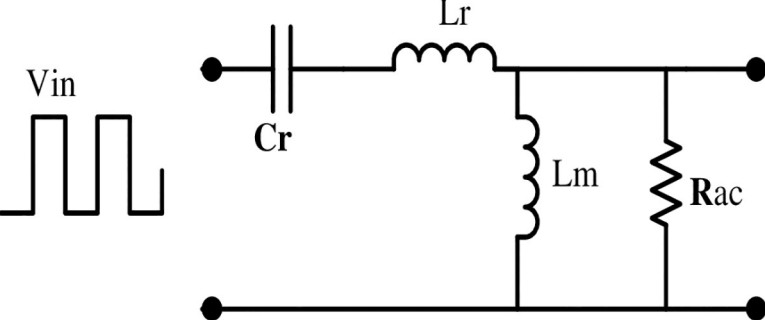

**Fig 5. Resonant equivalent circuit.**

The resonant frequency $f_r$ and magnetizing frequency $f_m$ can be expressed as by the following relations:

$$fr = \frac{1}{2\pi\sqrt{Cr\ Lr}} \tag{2}$$

$$fm = \frac{1}{2\pi\sqrt{Cr\ (Lr+Lm)}} \tag{3}$$

Eq 4 expresses the quality factor Q of the system:

$$Q = \sqrt{\frac{L_r}{C_r}}\frac{1}{R_e} \tag{4}$$

The leakage inductor on the secondary side, has a minute value, which could be ignored, so that the voltage gain is calculated using Eq 5, where $V_o$ is the output voltage, $A = L_m/L_r$ is the inductance ratio and $f_n = f_s/f_r$. With the higher value of $A$, the value of the Maximum Voltage Gain, the value of turning point frequency and the variation in the bus voltage from the full to light-load state, will be smaller. In addition, for constant value of $f_r$, the range of $f_s$ will be wider.

$$M = \left|\frac{\frac{4n\ V_o}{\pi}}{\frac{2}{\pi}V_{bus}}\right| = \frac{A}{\sqrt{\left(1+A-\frac{1}{f_n^2}\right)^2 + Q^2A^2\left(f_n - \frac{1}{f_n}\right)^2}} \tag{5}$$

The average voltage contained by resonant capacitor is half of the magnitude of bus voltage, while the maximum voltage of resonant capacitor $V_{c\ (max)}$ can't be greater than the bus voltage value as shown in (6). The relation as shown in Eq (7) can calculate the minimum capacitance value of resonant capacitor Cr(min). Here $I_o$ is the current through the load and $T_{max}$ is the maximum switching time period.

$$V_{c\ max} = nV_o + \frac{I_o T_{max}}{4nC_r} = V_{bus} \tag{6}$$

$$C_{r\ min} = \frac{I_0 T_{max}}{4n(V_{bus} - nV_o)} \tag{7}$$

## High voltage transformer

The high voltage transformer (HVT) can be elaborated in Fig 6, which is used in the proposed circuit design. The (HVT) can be designed by using all the magnetic components including magnetizing inductors and leakage/resonant incorporated in it, but it became more complex. In order to avoid the complexity of the HVT structure, complicated an improper approach of magnetic components is not used in proposed topology. Therefore, in the proposed topology separate Lm and Lr are used. The high voltage transformer (HVT) can be used to keep low leakage inductance with independent leakage. So, the secondary windings are wound on top and primary are wound below of them. Such type of structure windings results in uniform coupling between the secondary and the primary windings and low leakage inductances.

## Simulation results

To formalize the achievable impacts of the proposed DC power supply in producing high-voltage, a simulation based on OrCAD PSpice testing was conducted. The simulation model have the following specifications: Vin = 220 V, Lf = 5mH, Lb = 200uH, C1 = C2 = 330nF, Cbus = 10uF, Lm = 100uH, Cr = 30nF, Lr = 20μH, and Co = 10nF. The proposed topology was simulated at maximum switching frequency of 50kHz in order to produce maximum output voltage. The Figs 7, 8 and 9 given below show the waveforms of output voltage, output current that is 2kv and input source voltage.

## Power factor correction experimental waveforms

The Fig 10 given below show the input voltage and input current waveforms of the proposed power supply at full load. The power supply is operating at switching frequency of 50KHz,

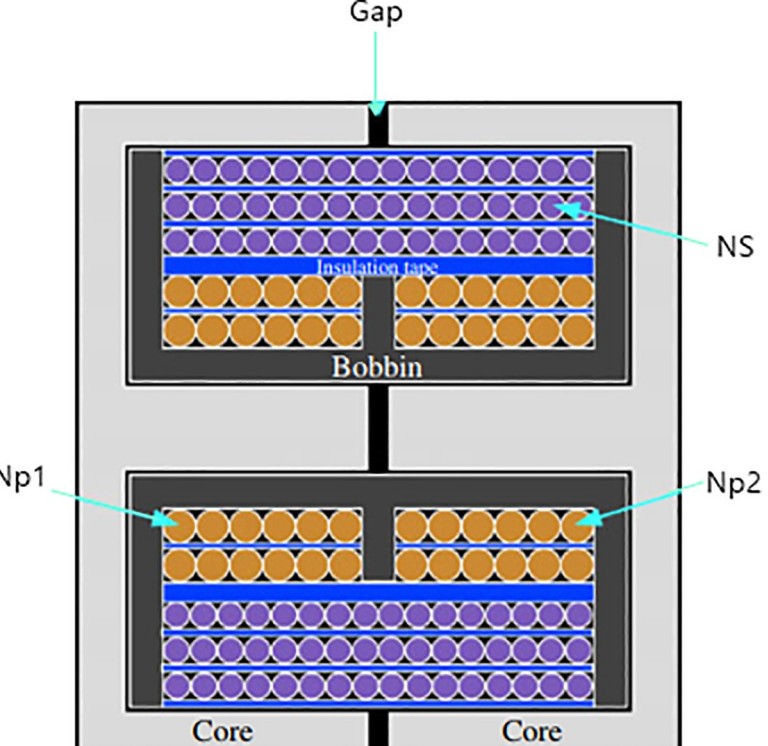

**Fig 6. High voltage transformer.**

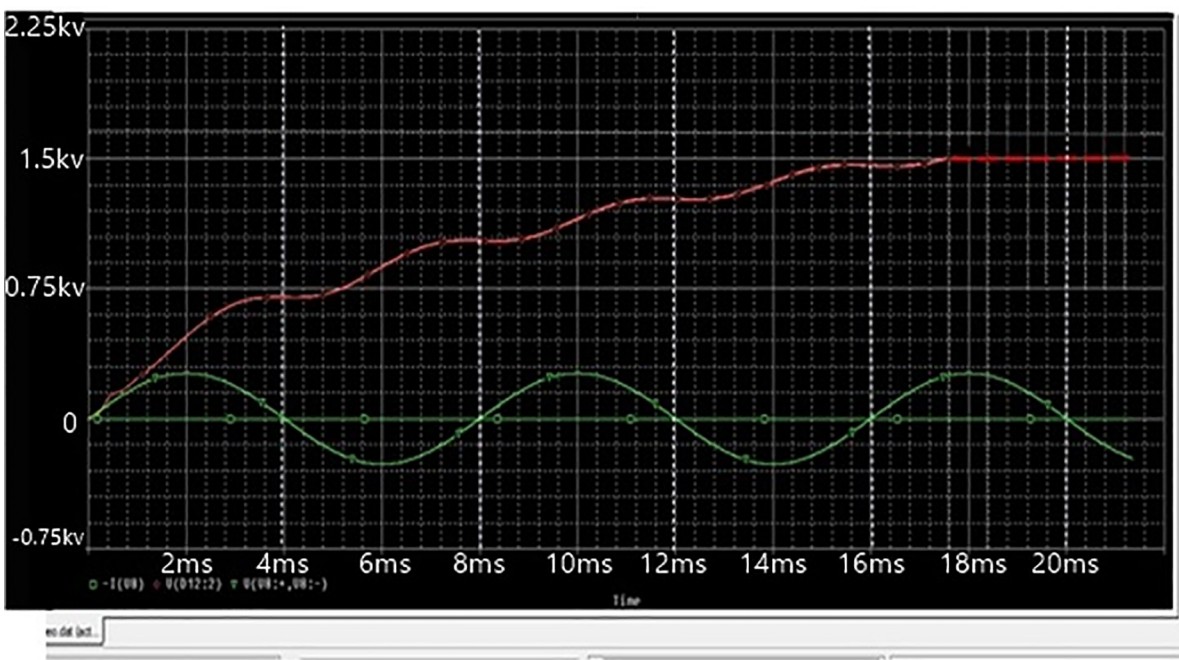

**Fig 7. Output and input source voltage waveforms of power supply.**

220V input voltage, transformer turn ratio 65:175 and 210.0W output power. As it is very clear, that the input voltage and input current have the sinusoidal shapes and are having no phase difference. Therefore, the collective harmonics distortion of current is less than 4% and power factor is greater than 0.98.

## Experimental waveforms of output voltage

By varying the switching frequency of the power supply, the proposed power supply's output can be controlled. The power supply is operated in between switching frequency range (fs,

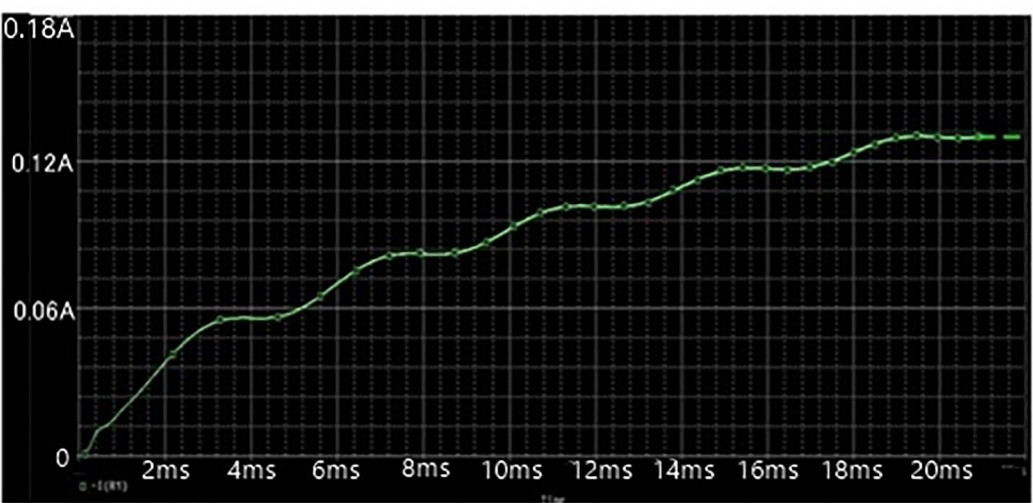

**Fig 8. Output current waveform.**

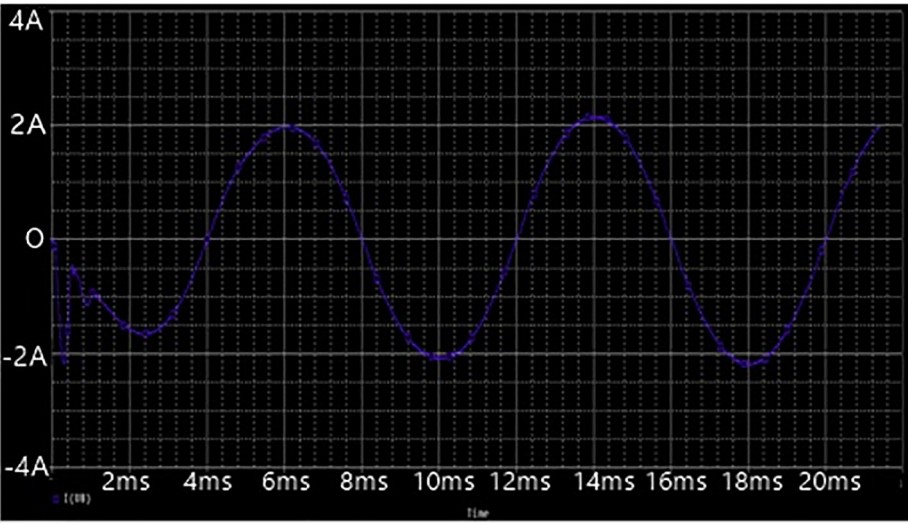

**Fig 9. Input source current waveform.**

max) would that it can't exceed the maximum switching frequency. To exhibit the effectiveness of the experimental proposed power, supply the experimental results only for fs = 50 kHz are presented. The proposed power supply is very significant for the high-voltage applications. The power and voltage of the power supply can be increased by increasing the turn ratio of HVT and to increase the input DC voltage. However, for increasing the turn ratio of HVT, the transformer would have to be reconstructed with a sufficient distance between primary and secondary windings, but occurrence of an increase in leakage inductance will also be a limitation here.

As "power-Mbreakn" can operate well up to 50 kHz, so by employing the "power-Mbreakn", there would here will problem related to frequency limitation if "power-Mbreakn"

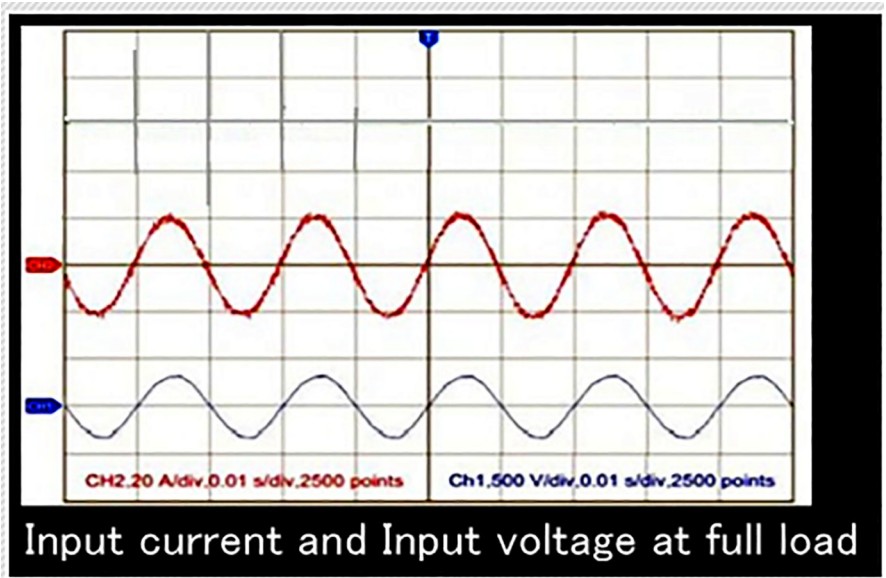

**Fig 10. Input current and voltage at full load.**

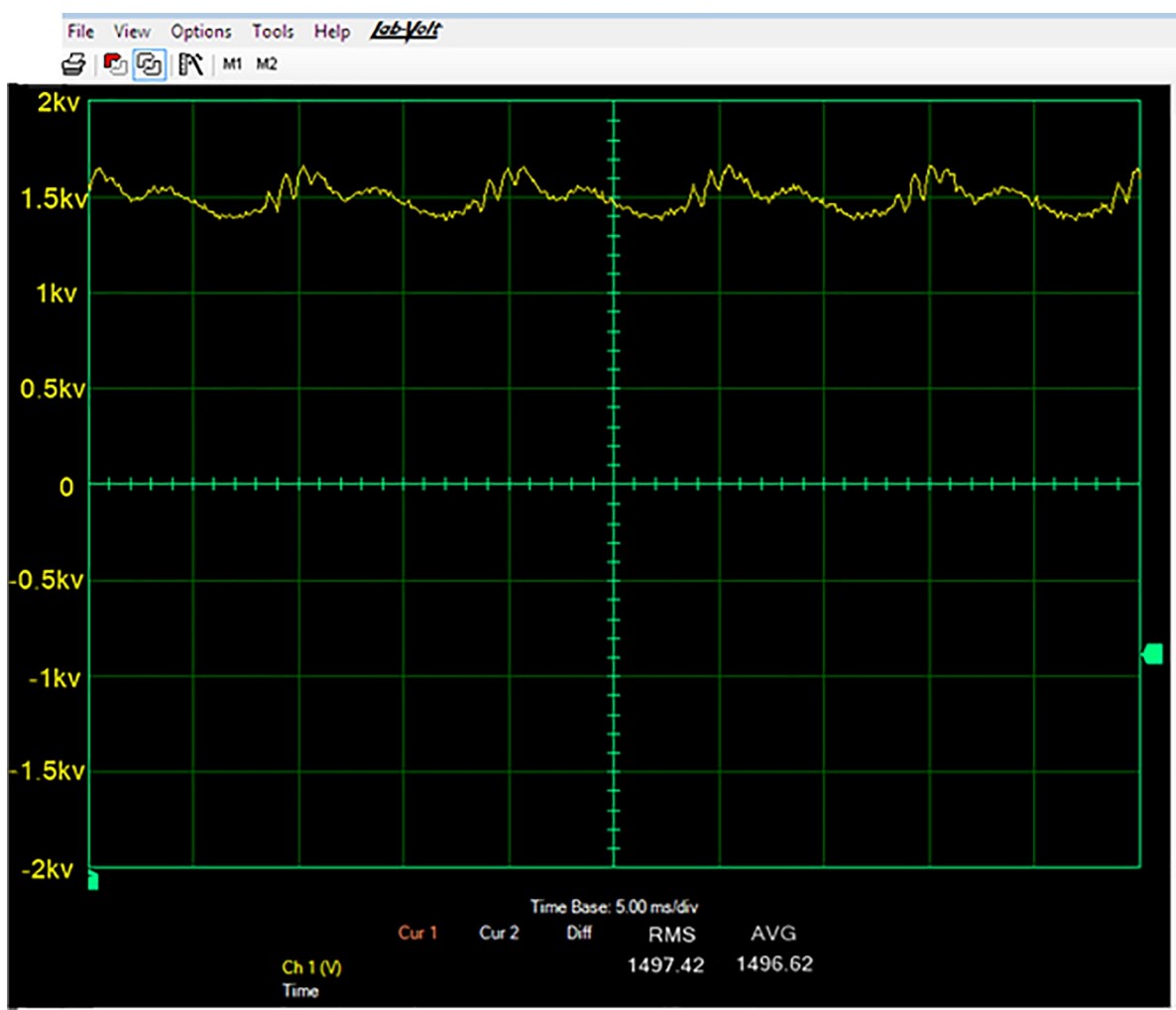

**Fig 11. Output voltage of proposed topology.**

are employed. It was observed that due to ZCS operation of power supply, the power MOS-FETs "power-Mbreakn" have good operation in the proposed topology. The designed prototype of the proposed power supply was tested experimentally for steady-state operation and shown in Fig 11 below.

## Hardware of proposed topology

A prototype system as shown in Fig 12 has been build, in order to validate the proposed concept of high voltage power supply. The experimental setup is composed of an input rectifier (D1- D4) with two filter capacitors (C1 & C2), two inductors (Lf & Lb), two diodes (D9 & D10), a bus capacitor (Cbus), a high-frequency inverter with LLC resonant tank, an HVT with high frequency, a high voltage secondary rectifier with an output filter capacitor co and an output resistive load at the end. In order to produce driving pulses for power switches "IOR2112 Controller" is employed in the proposed topology and therefore, a separate circuit containing "IOR2112 Controller" is built to generate driving pulses of 50kHz for MOSFETs.

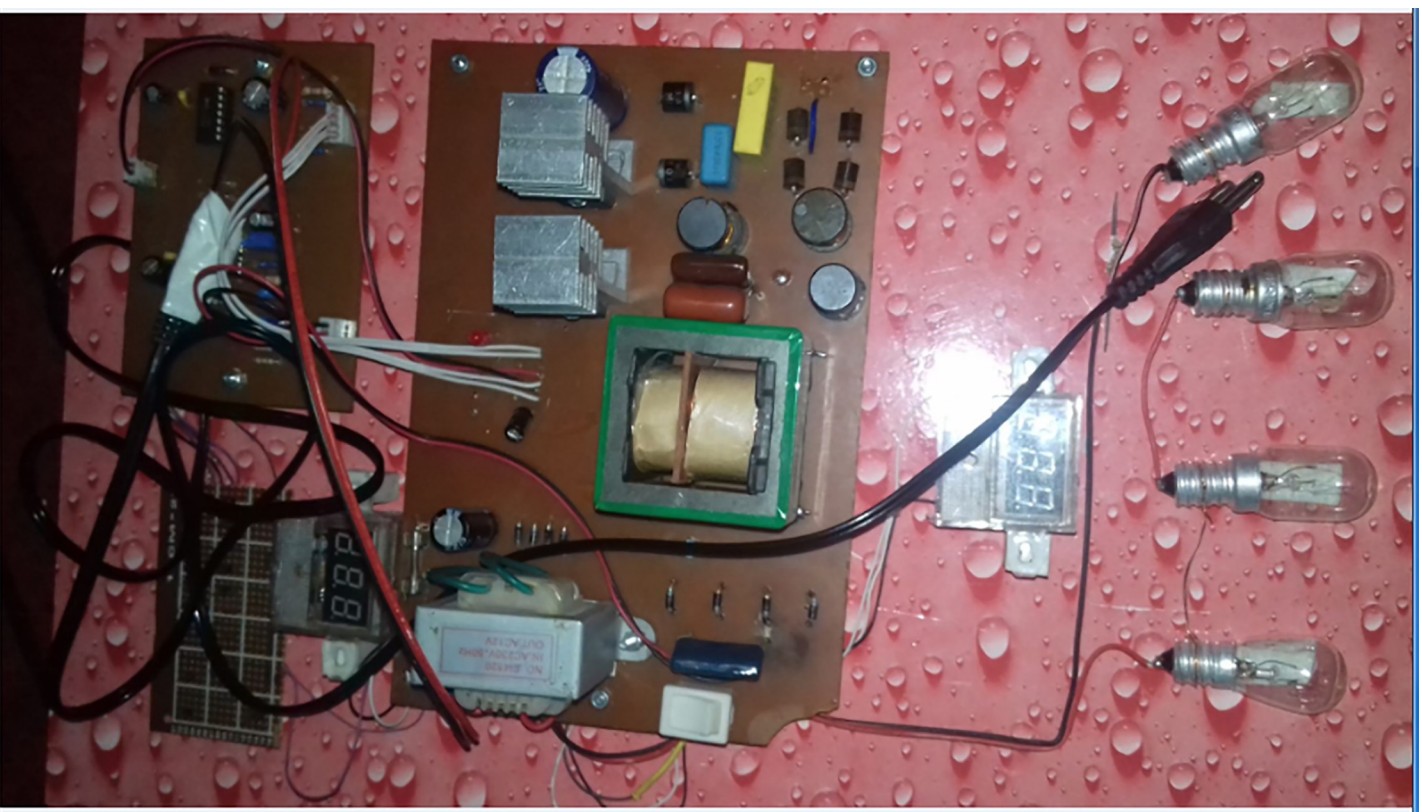

**Fig 12. Complete hardware.**

## Conclusion

In the proposed paper, a new topology based on LLC resonant converter with power factor correction for higher voltage applications is introduced. The proposed topology performs input power factor correction (PFC) at high output voltage. The Cbus capacitor generate 400v with input power factor 0.98. The output voltage of proposed system can be controlled by changing the switching frequency applied at the gate terminal of the power MOSFETs. To validate the performance of proposed power supply, which is tested by simulation by ORcAD PSpice software. It is also proven by running the simulation from the simulation results that, this proposed topology is to generate high voltage very efficiently. As both power switches are turned-on and turned-off under soft switching condition, therefore, no switching losses occurs in proposed topology.

In addition, the proposed topology is simple to work, small and light in size and weight respectively. Therefore, the proposed high voltage DC power supply with power factor correction is verified as an alternative for high voltage applications.

## Supporting information

**S1 File.**
(DOCX)

**S2 File.**
(DOCX)

## Author Contributions

**Conceptualization:** Fiaz Ahmad.

**Formal analysis:** Saeed Nawaz.

**Project administration:** Muhammad Abid.

**Resources:** Farman Ullah.

**Software:** Usman Habib.

**Validation:** Mohsin Iqbal, Ajmal Farooq.

**Writing – original draft:** Usman Habib.

**Writing – review & editing:** Fiaz Ahmad.

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
