## [Decision Letter · Decision Letter 0]

5 Aug 2020

PONE-D-20-11491

High voltage DC power supply with power factor correction based on LLC Resonant Converter

PLOS ONE

Dear Dr. Iqbal,

Thank you for submitting your manuscript to PLOS ONE. After careful consideration, we feel that it has merit but does not fully meet PLOS ONE’s publication criteria as it currently stands. Therefore, we invite you to submit a revised version of the manuscript that addresses the points raised during the review process.

ACADEMIC EDITOR:

A minor revision is needed for this paper. 

We look forward to receiving your revised manuscript.

Kind regards,

Long Wang, Ph.D.

Academic Editor

PLOS ONE

Journal Requirements:

2.We suggest you thoroughly copyedit your manuscript for language usage, spelling, and grammar. If you do not know anyone who can help you do this, you may wish to consider employing a professional scientific editing service.  

3.In your Data Availability statement, you have not specified where the minimal data set underlying the results described in your manuscript can be found. PLOS defines a study's minimal data set as the underlying data used to reach the conclusions drawn in the manuscript and any additional data required to replicate the reported study findings in their entirety. All PLOS journals require that the minimal data set be made fully available. For more information about our data policy, please see http://journals.plos.org/plosone/s/data-availability.

4.Thank you for stating the following in your Competing Interests section: 

[I have read the journal's policy and the authors of this manuscript have the following competing interests].

5. Please ensure that you refer to Figure 5 in your text as, if accepted, production will need this reference to link the reader to the figure.

6.Please upload a copy of Figure 12, to which you refer in your text on page 20. If the figure is no longer to be included as part of the submission please remove all reference to it within the text.

Reviewers' comments:

Reviewer's Responses to Questions

**Comments to the Author**

1. Is the manuscript technically sound, and do the data support the conclusions?

Reviewer #1: Yes

2. Has the statistical analysis been performed appropriately and rigorously? 

Reviewer #1: Yes

3. Have the authors made all data underlying the findings in their manuscript fully available?

Reviewer #1: Yes

4. Is the manuscript presented in an intelligible fashion and written in standard English?

Reviewer #1: Yes

5. Review Comments to the Author

Reviewer #1: The authors proposed a novel topology based on LLC resonant converter with power factor correction for higher voltage applications. The reviewer has the following comments:

1. the Figure 1 is fuzzy.

2. Authors should include references from recent years.

6. PLOS authors have the option to publish the peer review history of their article (what does this mean?). If published, this will include your full peer review and any attached files.

Reviewer #1: No

---

## [Author Response · Author response to Decision Letter 0]

23 Aug 2020

Thanks for giving us an opportunity to submit a revised version of manuscript entitled “High voltage DC power supply with power factor correction based on LLC Resonant Converter” for possible publication in your esteemed journal. We appreciate the time and effort put by you and reviewer(s). We have incorporated the changes required and these are highlighted in red color within the manuscript. Please see below in blue, point by point response to the reviewer(s) comments.

Reviewer(s) Comments to Authors:

Reviewer 1:

1. The Figure 1 is fuzzy.

Author(s) Response: Thank you for pointing out this, Fig. 1 has been redrawn and its quality is improved in revised manuscript

2. Authors should include references from recent years.

Author(s) Response: Most of the references have been updated with recent ones in the revised manuscript

---

## [Decision Letter · Decision Letter 1]

28 Aug 2020

High voltage DC power supply with power factor correction based on LLC Resonant Converter

PONE-D-20-11491R1

Dear Dr. Iqbal,

We’re pleased to inform you that your manuscript has been judged scientifically suitable for publication and will be formally accepted for publication once it meets all outstanding technical requirements.

Kind regards,

Long Wang, Ph.D.

Academic Editor

PLOS ONE

Additional Editor Comments (optional):

Reviewers' comments:

Reviewer's Responses to Questions

**Comments to the Author**

1. If the authors have adequately addressed your comments raised in a previous round of review and you feel that this manuscript is now acceptable for publication, you may indicate that here to bypass the “Comments to the Author” section, enter your conflict of interest statement in the “Confidential to Editor” section, and submit your "Accept" recommendation.

Reviewer #1: All comments have been addressed

2. Is the manuscript technically sound, and do the data support the conclusions?

Reviewer #1: Yes

3. Has the statistical analysis been performed appropriately and rigorously? 

Reviewer #1: Yes

4. Have the authors made all data underlying the findings in their manuscript fully available?

Reviewer #1: Yes

5. Is the manuscript presented in an intelligible fashion and written in standard English?

Reviewer #1: Yes

6. Review Comments to the Author

Reviewer #1: This paper is revised well and all modifications suggested by the reviewers were made.

7. PLOS authors have the option to publish the peer review history of their article (what does this mean?). If published, this will include your full peer review and any attached files.

Reviewer #1: No

---

## [Editor Report · Acceptance letter]

8 Sep 2020

PONE-D-20-11491R1 

High voltage DC power supply with power factor correction based on LLC Resonant Converter 

Dear Dr. Iqbal:

I'm pleased to inform you that your manuscript has been deemed suitable for publication in PLOS ONE. Congratulations! Your manuscript is now with our production department. 

Kind regards, 

on behalf of

Dr. Long Wang 

Academic Editor

PLOS ONE